# High-Efficiency Inscription of Fiber Bragg Grating Array with High-Energy Nanosecond-Pulsed Laser Talbot Interferometer

**DOI:** 10.3390/s20154307

**Published:** 2020-08-01

**Authors:** Zhe Zhang, Baijie Xu, Jun He, Maoxiang Hou, Weijia Bao, Yiping Wang

**Affiliations:** 1Guangdong and Hong Kong Joint Research Centre for Optical Fiber Sensors, College of Physics and Optoelectronic Engineering, Shenzhen University, Shenzhen 518060, China; 2150190115@email.szu.edu.cn (Z.Z.); 1900453046@email.szu.edu.cn (B.X.); maoxiangh@gdut.edu.cn (M.H.); wjbao@szu.edu.cn (W.B.); ypwang@szu.edu.cn (Y.W.); 2Guangdong Laboratory of Artificial Intelligence and Digital Economy (SZ), Shenzhen University, Shenzhen 518060, China; 3Key Laboratory of Optoelectronic Devices and Systems of Ministry of Education and Guangdong Province, Shenzhen University, Shenzhen 518060, China; 4School of Electromechanical Engineering, Guangdong University of Technology, Guangzhou 510006, China

**Keywords:** fiber Bragg grating array, laser Talbot interferometer, mass production

## Abstract

A high-energy nanosecond-pulsed ultraviolet (UV) laser Talbot interferometer for high-efficiency, mass production of fiber Bragg grating (FBG) array was experimentally demonstrated. High-quality FBG arrays were successfully inscribed in both H_2_-free and H_2_-loaded standard single-mode fibers (SMFs) with high inscription efficiency and excellent reproducibility. Compared with the femtosecond pulse that had a coherent length of several tens of micrometers, a longer coherent length (~10 mm) of the employed laser rendered a wider FBG wavelength versatility over 700 nm band (1200–1900 nm) without the need for optical path difference (OPD) compensation. Dense FBG array with center wavelength separation of ~0.4 nm was achieved and more than 1750 FBGs with separated center wavelength could be inscribed in a single H_2_-free or H_2_-loaded SMF in theory, which is promising for mass production of FBG arrays in industry. Moreover, precise focusing of laser beam was superfluous for the proposed system due to the high energy density of pulse. The proposed FBG inscription system was promising for industrialization production of dense FBG arrays.

## 1. Introduction

Over decades, tremendous progress has been made in fiber Bragg gratings (FBGs) due to its extensive applications in optical fiber lasers [1,2,3], optical fiber communications [4,5,6] and single-point [7,8,9] or quasi-distributed [10,11] optical fiber sensing fields. For more recently, FBG arrays are widely employed in quasi-distributed sensing networks for long distance, online health monitoring of large structures [12,13]. For instance, in 2012, Zhang et al. [14] demonstrated a quasi-distributed optical fiber sensing system based on identical weakly reflective FBG array and optical wavelength time–domain reflection technique (OWTDR). The sensing system achieved a large multiplexing capacity up to over 1000 in theory. In 2017, Gui et al. [15] developed a dense weak-reflection FBG array with identical FBG central wavelength for high spatial resolution distributed sensing applications. They demonstrated a high spatial resolution over 6680 FBGs along a 10-m-long fiber. More recently, Zhang et al. [16] proposed a hybrid wavelength and frequency division multiplexing (FDM) architecture based on cascading two identical weakly reflective FBGs for distribute pressure measurements. As a key component in quasi-distributed optical fiber sensing networks, the inscription and integration of FBGs with separated center wavelength can improve the multiplexing capacity of the sensing network significantly.

Tremendous methods have been developed for the inscription of FBG array with separated wavelength, such as femtosecond laser point-by-point [17], line-by-line [18] and laser Talbot interferometer methods [19]. The femtosecond laser point-by-point and line-by-line FBG inscription approaches are flexible to design the period of FBGs [20]. However, these methods are heavily depended on ultrafast lasers and the laser beam needs to be precisely focus onto the fiber core due to the nonlinear multiphoton absorption mechanism and threshold effects of ultrafast laser [21]. Thus, an expensive femtosecond laser, high-quality microscope and nano-precision positioning stage are necessary for the FBG inscription, and the inscription efficiency and spectra quality of inscribed FBG are poor with large insertion loss, especially at shortwave region. Alternatively, laser Talbot interferometric FBG inscriptions system exhibits great flexibility, large wavelength versatility, low cost and mass production abilities. The period of interference pattern can be tuned by rotating the rotation mirrors and thus producing FBGs with different resonant wavelength. Wang et al. proposed a continuous-wave UV-laser Talbot interferometer for FBG inscription in 2001 [22]. They employed a 40-mw continuous-wave frequency-doubled argon-ion laser at 244 nm and a single phase mask as a beam splitter. The ultralong coherent length of the laser source renders the system flexibility in inscribing FBGs with central wavelength ranging from 600–1300 nm. However, the continuous-wave UV laser has low peak power and thus the FBG inscription system shows low FBG inscription efficiency, especially for low-photosensitivity or H_2_-free optical fibers. In 2008, femtosecond-laser Talbot interferometer for FBG inscriptions were first, experimentally demonstrated by M. Becker et al. [23]. The femtosecond laser has powerful processing ability for a variety of optical fibers [8,19]. The combination of femtosecond laser source and Talbot interferometer not only gives access to the fabrication of Bragg gratings in new types of materials, but also allows to keep high flexibility in choosing the Bragg wavelength. However, due to the poor spatial and temporal coherence properties of the femtosecond laser source, specific limits and tolerances in the interferometric setup must be considered. The environment disturbance, fiber displacement and off-axis position affect the interference pattern significantly. After changing the laser incident angle by rotating the two rotation mirrors, the optical path difference (OPD) changed [23], which can result in the degeneration of interference pattern. Optical delay line is typically employed to compensate the OPD between the two beams and the achievable FBG wavelength width is limit to several tens of nanometer. In addition, the demand for precise focus of femtosecond laser onto the fiber core imposed another formidable hinder. As such, femtosecond laser Talbot interferometric FBG inscription system cannot be employed for mass production of FBG arrays, especially for industrial applications.

In this study, we report on a reliable laser Talbot interferometer FBG inscription system, which overcome the deficiency of continuous-wave UV-laser and femtosecond laser Talbot interferometer FBG inscription system. A high-energy quadruple-frequency Nd–YAG nanosecond-pulsed UV laser that has a coherent length of ~10 mm is employed as the laser source and high-quality FBG and FBG arrays are successfully inscribed in both H_2_-free and H_2_-loaded SMFs with high efficiency and excellent reproducibility. The center wavelength of inscribed FBGs can be tuned continuously from 1200 to 1900 nm without any compensation for the OPD of the two beams and more than 1750 FBGs with separated center wavelength can be inscribed in a single fiber. Such properties of the proposed FBG inscription system make it a good candidate for mass production of FBG arrays for industry applications.

## 2. Sensor Fabrication and Working Principle

The proposed laser Talbot interferometric FBG inscription setup is schematically illustrated in Figure 1a. First, the fourth harmonic of a 10 ns pulsed Nd–YAG laser (Quantum-Ray Lab-170, Spectra Physics, CA, USA) with center wavelength of 266 nm and repetition rate of 10 Hz passes through a half wave plate (WPH05M-266, Thorlabs, Newton, NJ, USA) followed by a Glan Polarizer (GLPB2-06-29SN-2/3, Sigma, Kanagawa, Japan). The combination of a half wave plate and Glan Polarizer functions as an energy regulator for the laser beam. With the energy regulator, the pulse energy could be tuned continuously from 100 to 0.1 mJ. Then the laser beam was focused by a long-focus cylindrical lens (CSX300AR.10, Newport, RI, USA) through a uniform phase mask (On-tech UM-1065-20*10, OVLINK, Wuhan, China) and a pair of coaxial rotation reflectors onto the fiber (SMF-28e, Corning, Midland, NC, USA). The phase mask functioned as a beam splitter and the 0-order diffraction beam of the phase mask was blocked. The divisive ±1 order diffraction beams were reflected by a pair of electrical coaxial rotation reflectors and recombined at a determined position, forming a space interference pattern. The cylindrical lens had a focal length of ~300 mm and was fixed to a motorized stage (M-ILS100CCL, Newport, RI, USA) to adjust the focusing distance.

Compared with the femtosecond laser Talbot interferometer FBG inscription system, where the laser beam needs to be precisely focused onto the fiber core due to the nonlinear absorption mechanism and threshold effects of femtosecond laser micromachining, there was no demand for precise focusing in the proposed FBG inscription system. This property could be attributed to the high energy dense of the employed laser source and that the photosensitive region of the fiber was merely limited to the Ge-doped fiber core. The employed laser (Quantum-Ray Lab-170, Spectra Physics, CA, USA) was spatial single mode that had a coherent length of ~10 mm, which greatly reduced the difficulty for balancing the OPD of the two beam that interfered. The employed uniform phase mask (On-tech UM-1065-20*10, OVLINK, Wuhan, China) had a period of 1065 nm and ±1 order diffraction efficiency of ~27%. The fiber was held by a pair of fiber holders and the fiber holders were fixed to a motorized stage. The motorized stage had a large trip range of ~300 mm, which endowed the system capacity for inscribing FBGs with a wide wavelength versatility.

The principle for inscribing FBG with different central wavelength is schematically illustrated in Figure 1b. The initial angle of the rotation mirror is 0 and the angle α of ±1 order beams diffracted from the phase mask can be calculated by:(1)$$\alpha =\arcsin(\frac{\lambda_{laser}}{\Lambda^{\prime }}),$$
where *λ_laser_* is the incident wavelength, Λ′ is the period of phase mask. The initial interference angle θ is equal to the angle of ±1 order diffraction beams α (i.e., θ = *α*). The period Λ of the interference pattern (i.e., the grating pitch of FBG) can be expressed as:(2)$$\Lambda =\frac{\lambda_{laser}}{2\sin\theta },$$then the central wavelength of FBG can be expressed as:(3)$$\lambda_{Bragg}=2n_{eff}\Lambda =\frac{n_{eff}\lambda_{laser}}{\sin\theta },$$
where *λ*_Bragg_ is the central wavelength of FBG and *n*_eff_ is the effective refractive index (RI) of the fiber mode. When rotating the mirror for an angle Φ, the angle of laser beam changed 2Φ, as is clearly shown in Figure 1b, the interference angle θ changed to θ’ (θ’ = θ + 2Φ). The central wavelength of FBG can be calculated by:(4)$$\lambda_{Bragg}=2n_{eff}\Lambda =\frac{n_{eff}\lambda_{laser}}{\sin\theta^{\prime }}=\frac{n_{eff}\lambda_{laser}}{\sin(\theta +2\Phi )},$$
where *λ*_Bragg_ changes with the angle Φ. When the angle of rotation mirrors changed, the position of intersection point of the two beams changed accordingly. As is shown in Figure 1b, where the intersection point of the two beams changed from A to A’. As a result, the fiber needs to be moved for a certain distance d_2_ to ensure the overlap between the fiber and the intersection point of the two laser beams and the cylindrical lens needs to be moved a distance d_1_ accordingly. It is worth noting that the employed cylindrical lens has a long focal length of ~300 mm and the photosensitive areas of fiber is limited to the fiber core. As such, there is no need for precisely focusing of laser beam due to the single photon linear absorption mechanism of the Ge ion-doped fiber core to the UV illumination. This schema renders the FBG inscription system much more flexibility and practicability.

The relationships between the rotation angle Φ of the mirrors and the interference angle θ’, central wavelength λ_FBG_ of FBG are calculated according to Equation (4) and are listed in Table 1. Herein, the effective RI of fiber mode is set as 1.446, the incident wavelength is 266 nm and the ±1 order diffraction angle α of the phase mask is calculated to be 14.46° according to Equation (1).

The spatial energy density of the focused laser beam is estimated by employing a Gaussian optics model. The beam diameter *W*_0_ (1/e of the maximum) in front of the cylindrical lens (CL) is measured to be ~6 mm. As is illustrated in Figure 1c, the beam diameter W_Z_ after focusing by the CL can be expressed as:(5)$$W_{Z}\approx \frac{4\lambda_{0}f_{L}}{\pi W_{0}},$$
where *λ*_0_ and *f*_L_ denote the laser wavelength and the focal length of CL, respectively. The focal depth *L*_DOF_ is twice the Rayleigh length *Z*_R_ and can be expressed as:(6)$$L_{DOF}=2Z_{R}\approx \frac{8\lambda_{0}f_{L}{}^{2}}{\pi W_{0}{}^{2}},$$
in our case, the W_Z_ and *L*_DOF_ are calculated to be ~17 and 213 µm, respectively. The area of the focusing region *A*_Z_ can then be expressed as:(7)$$A_{Z}=W_{Z}\cdot W_{Z}\approx \frac{4}{\pi }\lambda_{0}f_{L},$$As such, the energy density *D* of the focus can be estimated to be:(8)$$D=\frac{E}{A_{Z}},$$
where *E* is the pulse energy. Considering that the pulse energy of the employed 266 nm laser can be tuned from 100 to 0.1 mJ continuously by the energy regulator, the maximum and minimum energy density *D* can be calculated to be ~100 J/cm^2^ and 100 mJ/cm^2^, respectively. It is worth noting that, due to the long focal length of the CL (~300 mm), it is difficult to precisely focusing, and, namely, a little defocusing is inevitable. Thus, the actual energy density that applied to the fiber is a little bit lower than the calculated value above.

## 3. Inscription of Various FBG Arrays

The inscription of various FBG arrays was experimentally carried out. The FBG inscription efficiency in H_2_-free Ge-doped SMF was improved significantly with the proposed inscription system, which was promising for mass production of weak-reflection FBG array for quasi-distributed fiber-optic sensing applications. The inscription of FBG array in H_2_-loaded SMF was also studied and FBG array with wavelength versatile over 1200–1900 nm and wavelength separation of ~0.4 nm in a single optical fiber was experimentally demonstrated.

### 3.1. FBG Array Inscribed in H_2_-Free SMFs

A length of standard SMF (SMF-28e, Corning, Midland, NC, USA) with Ge-doped core was employed and the coating of the fiber was first stripped away, followed by alcohol cleaning. Then, the prepared bare fiber was fixed to the motorized stage by a pair of fiber holders. The pulse energy was tuned to ~5 mJ by rotating the half wave plate. The angle of rotation mirrors was first set to 0 and the fiber was moved by the motorized stage to a position where the two beams combined. Figure 2a shows the reflection spectrum evolution of the FBG inscribed in a H_2_-free SMF. As is clearly shown in Figure 2a, after an exposure of ~10 s, a weak peak at ~1543.6 nm appeared in the reflection spectrum. The weak reflection peak grew gradually from ~−80 dBm to −65 dBm and the signal-to-noise ratio (SNR) reached ~15 dB after a laser illumination time of ~30 s. Then the reflection peak grew slowly, which may be attributed to the weak photosensitivity of the H_2_-free Ge-doped fiber core and the saturation of refractive index modulation. We measured the transmission spectrum of the inscribed FBG, as shown in Figure 2b, the FBG resonant dip in the transmission spectrum was shallow enough that it could hardly be observed. A conclusion can be made that the proposed Talbot interferometer setup has a relatively high FBG inscription efficiency for H_2_-free Ge-doped SMFs while the transmission spectrum of the inscribed FBG indicated an ultralow reflectivity.

FBG arrays with separated wavelength were also inscribed in a single H_2_-free SMF by changing the angle of the rotation mirrors together with moving the fiber to the intersection of the two beams that combined. After the inscription of one FBG, the laser beam was blocked and the mirrors were rotated for a determined angle Φ, followed by moving the fiber for a certain distance d_2_. Then, the laser blocker was removed, starting the inscription of the next FBG. The axial distance between the adjacent two FBGs was ~1.1 cm and the grating length was estimated to be ~6 mm according to the bandwidth of the inscribed FBG. Figure 3a–f shows the reflection spectra evolution of the FBG arrays during inscriptions. The exposure time for each FBG was ~10 s, and the pulse energy was tuned to ~8 mJ. The 23 weak-reflection FBGs with central wavelength ranging from 1500 to 1600 nm and SNR higher than 10 dB were inscribed in a single H_2_-free SMF successfully.

To improve the FBG multiplexing capacity within a single fiber in a limited bandwidth, the wavelength separation of two adjacent FBGs need to be narrowed. This could be achieved by reducing the rotation mirror angle interval during the inscriptions of each two adjacent FBGs. However, too small wavelength interval may result in a wavelength overlap between the two adjacent FBGs. This could be attributed to the discrimination of axial tensile stress applied to the fiber by the fiber holders during the inscription of each FBG. To be specific, after the inscription of one FBG, the fiber was released from the two fiber holders, and was moved a distance (~11 mm) along the fiber axial direction. Then the fiber was held by the fiber holders again, for the inscription of next FBG. As such, a minor difference in the tensile stress may result in an overlap of adjacent two FBG peaks due to the elastic–optic effect and the small wavelength interval of the adjacent two FBGs. However, this could be overcome by releasing one end of the fiber from the fiber holder and hanging a mass block on the fiber end, by which a similar tensile stress was ensured during inscription of each FBG [24]. Figure 4a shows the reflection spectrum of an FBG array inscribed with the common fiber holder approach. The achieved minimum peak separation in the experiments was ~1 nm and a narrower peak separation resulted in peak overlap. Figure 4b shows the reflection spectrum of an FBG array inscribed with a mass block hanging on one of the fiber end, the minimum peak separation achieved was ~0.4 nm. The employed hanging mass block was a spike, which was ~50 g in weight. The 3-dB bandwidth of the FBG reflection spectra was measured to be ~0.2 nm, corresponding to a grating length of ~8.3 mm [25].

In conclusion, owning to the high energy density of the employed nanosecond-pulsed UV laser source, weak-reflection FBG array could be easily inscribed in H_2_-free SMFs with high efficiency. By releasing one end of the fiber from the fiber holder and applying a constant weight to the fiber end, dense FBG array with a peak interval of ~0.4 nm was experimentally demonstrated in a single SMF.

### 3.2. FBG Array Inscribed in H_2_-Loaded SMFs

In order to improve the photosensitivity of the Ge-doped fiber core, a length of standard SMF was first sealed in a pure steel reaction still that filled with high pressure hydrogen. The applied pressure and temperature were 10 MPa and 80 °C, respectively. After maintaining the high-pressure and high-temperature environments for 7 days, the fiber was taken out from the still and then fixed to the motorized stage by the fiber holder for FBG inscription. The inscription processes were all the same as described above.

The pulse energy was tuned to ~5 mJ and the spectrum evolution was first observed in a reflection mode. As clearly shown in Figure 5a, the reflection spectrum grew rapidly and the reflection intensity reached ~−62 dBm, corresponding to a SNR of ~10 dB, after an exposure for ~10 s. A reflection intensity of ~−57 dBm and ~−51 dBm were recorded at an exposure time of ~20 s and ~30 s, respectively. We could clearly see that the growth speed of FBG reflection peak was much higher for a H_2_-loaded SMF than for a H_2_-free SMF. Then the spectrum evolution was observed in a transmission mode. Figure 5b shows the recorded transmission spectrum evolution. The blue solid line (T1) shows the transmission spectrum after an exposure of ~2 min, where the FBG resonant dip was ~−2 dB. With the dose of illumination increasing (~5 min exposure), the resonant dip of FBG grows from ~−2 dB to ~−10 dB (dash blue line (T2)), indicating an increasing refractive index modulation depth. A little “red shift” of the FBG resonant peak from ~1545.5 to 1545.8 nm was also observed, corresponding to the “DC” component of the refractive index modulation [25].

High-reflection FBG array with excellent spectral quality could be easily inscribed in a H_2_-loaded SMF. Figure 6a–c show the recorded spectra evolution of an FBG array consists of three FBGs with central wavelength of ~1502, 1546 and 1585 nm, respectively, in both reflection and transmission modes. The exposure time of each FBG was ~5 min and the resonant dips of the three FBGs in the transmission spectra were ~−7.5, −7.7 and −7.9 dB, respectively. The inscription of low-reflection FBG array was also experimentally studied. Figure 7a–f exhibit the reflection spectrum evolution of an FBG array consists of 20 weak-reflection FBGs inscribed in a single H_2_-loaded SMF. The exposure time for each FBG was ~5 s and the pulse energy was ~5 mJ. Considering the laser repetition rate of 10 Hz, each weak-reflection FBG in the FBG array was inscribed with an exposure of 50 pulsed. It was worth noting that, the SNR of each FBG was larger than 15 dB, which was high enough for the practical quasi-distributed sensing applications. Moreover, by increasing the pulse energy or by improving the photosensitivity of the fiber core, the FBG inscription efficiency could be improved further. However, one should note that pulse energy higher than 8 mJ (in our case) may break the fiber.

The minimum and maximum resonant wavelengths of an FBG array inscribed by the proposed Talbot interferometer without compensating of OPD were experimentally explored. From Table 1 and Equation (4), we could see that a larger rotation angle Φ was necessary for the inscription of FBG with a shorter or longer resonant wavelength. However, a larger rotation angle Φ brought about a serious problem, i.e., the OPD between the two interference beams were larger, which may result in the degeneration of interference pattern according to the detailed analyses presented in [23]. Figure 8 shows an inscribed FBG array with resonant wavelength ranging from ~1269.8 to ~1838.5 nm. The minimum and maximum central wavelength achieved without OPD compensation was ~1269.8 and 1838.5, respectively. The exposure time for the ten FBGs were 100, 80, 60, 40, 20, 40, 60, 80, 100 and 100 s—corresponding the resonant wavelength of 1269.8, 1327.7, 1391.7, 1460.2, 1538.6, 1627.1, 1675.4, 1729.3, 1780.7 and 1838.5 nm, respectively. It is worth noting that the FBG inscription efficiency declined significantly with the increasing of the rotation angle Φ. The reflection intensity and inscription efficiency reach maximum when the rotation angle Φ was 0. This agrees well with the analyses presented in [23]. This adverse impact was aggravated for a femtosecond laser Talbot interferometer FBG inscription system due to the ultrashort coherent length of femtosecond pulse (several tens micrometer).

Considering the wavelength versatility (1200–1900 nm) of the proposed FBG inscription system and the achieved minimum peak interval of ~0.4 nm, more than 1750 FBGs with separated central wavelength could be inscribed in a single SMF in theory. Here, two points are worth noting. First, the FBG inscription efficiency and the minimum or maximum resonant wavelength of FBG achieved were also affected by the emission spectrum of employed broadband source (BBS) and the transmission loss of the employed SMF. To be specific, the employed BBS was an amplified spontaneous emission white-light source, the light intensity at long wavelength or short wavelength was lower than that in the center of the emission spectrum. In addition, the employed SMF was optimized for the transmission of 1550 nm wavelength, light with a longer or shorter wavelength suffers a larger transmission loss. As such, the reflective intensity in short or long wavelength was already weak. Second, the proposed FBG inscription system could be improved further for fully automation online FBG inscriptions. By adding a motorized axial drag stage and linking all the motorized stage together by a LABVIEW program, fully automation, online inscriptions of FBG array in both H_2_-free and H_2_-loaded SMF could be achieved. This was of great significance in mass production and industrial application of FBG array in quasi-distributed optical fiber sensing networks.

## 4. Conclusions

In conclusion, a promising high-energy nanosecond-pulsed laser Talbot interferometer FBG inscription system was experimentally demonstrated. High FBG inscription efficiency and wide wavelength versatility (1200–1900 nm) were achieved without OPD compensation for the setup. The inscriptions of high-reflection FBG array in H_2_-loaded SMF and weak-reflection FBG array in both H_2_-loaded and H_2_-free SMF were experimentally demonstrated, respectively. A dense FBG array that had a peak interval of ~0.4 nm was achieved by applying a constant weight to one end of the fiber. More than 1750 FBGs with separated central wavelength can be inscribed in a single SMF. The proposed FBG inscription system has advantages of excellent flexibility, high inscription efficiency, which is a promising candidate for mass production and industry application of FBG arrays.

## Figures and Tables

**Figure 1 sensors-20-04307-f001:**
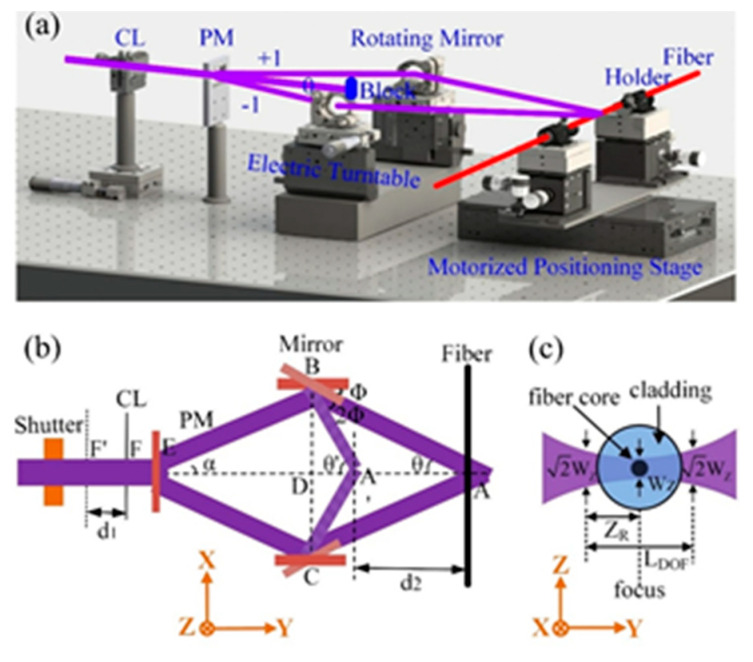
(**a**) Setup schematic and (**b**) optical path schematic of the proposed Talbot interferometer FBG inscription setup; (**c**) Gaussian optics model of the proposed Talbot interferometer; (CL—cylinder lens; PM—phase mask).

**Figure 2 sensors-20-04307-f002:**
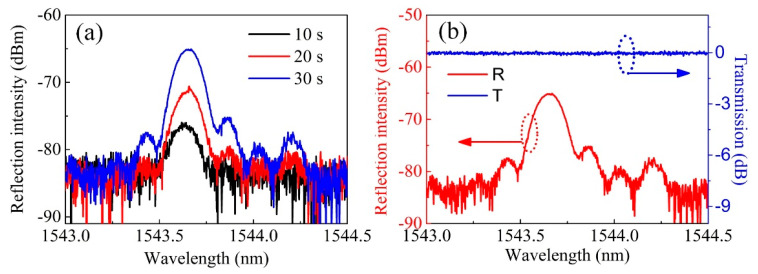
(**a**) Reflection spectrum evolution and (**b**) the ultimate reflection and transmission spectra of the FBG inscribed in a H_2_-free single-mode fiber (SMF).

**Figure 3 sensors-20-04307-f003:**
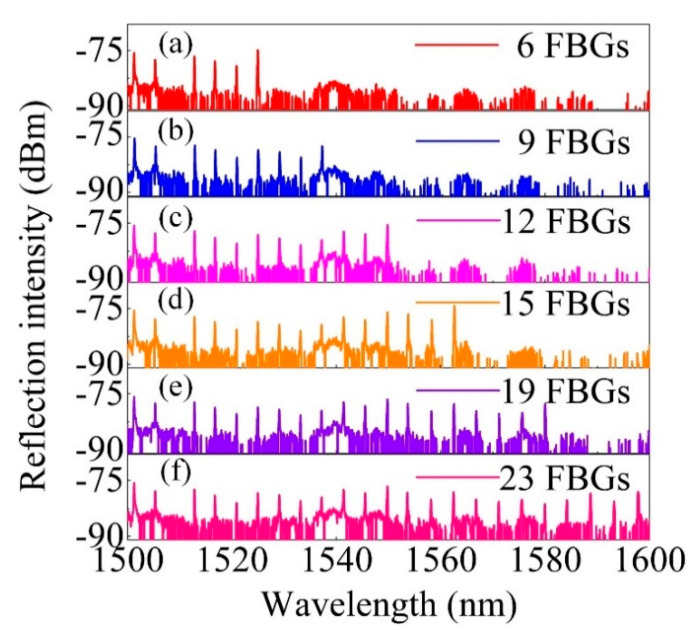
(**a**–**f**) Reflection spectrum evolution of an FBG array inscribed in a H_2_-free SMF.

**Figure 4 sensors-20-04307-f004:**
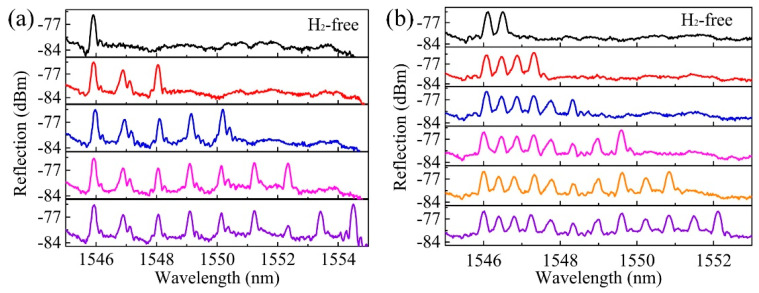
Reflection spectrum evolution of an FBG array inscribed with the (**a**) common fiber holder method and (**b**) with a mass block hanging on one end of the fiber. The peak separations are of ~1 nm and ~0.4 nm, respectively.

**Figure 5 sensors-20-04307-f005:**
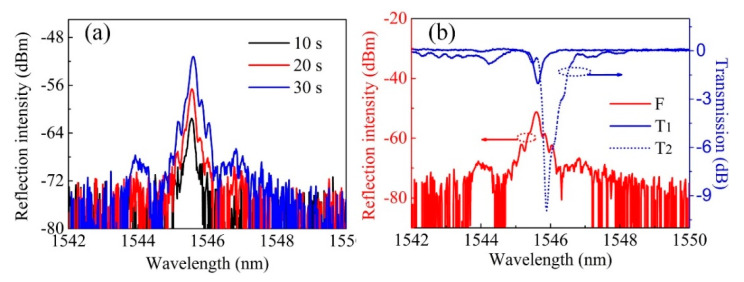
(**a**) Reflection and (**b**) transmission spectra evolutions of the FBG inscribed in a H_2_-loaded SMF.

**Figure 6 sensors-20-04307-f006:**
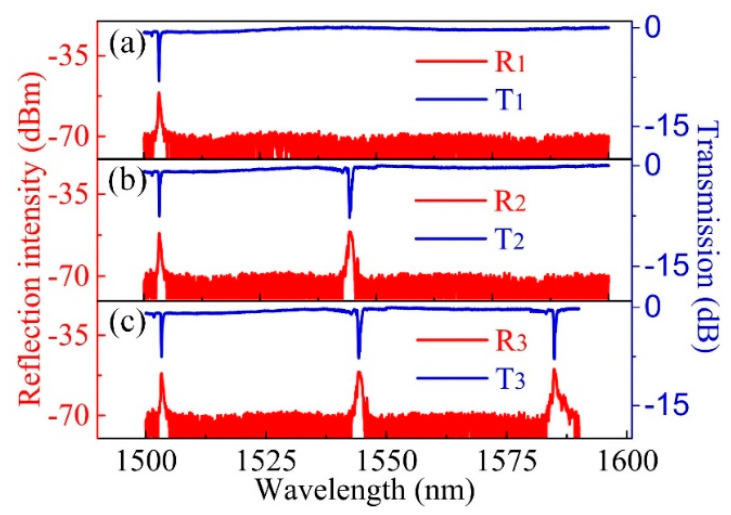
(**a**–**c**) Spectra evolution (red line: reflection; blue line: transmission) of an FBG array consists of three high-reflection FBGs. The resonant dips of the three FBGs are ~−7.5, −7.7 and −7.9 dB, respectively.

**Figure 7 sensors-20-04307-f007:**
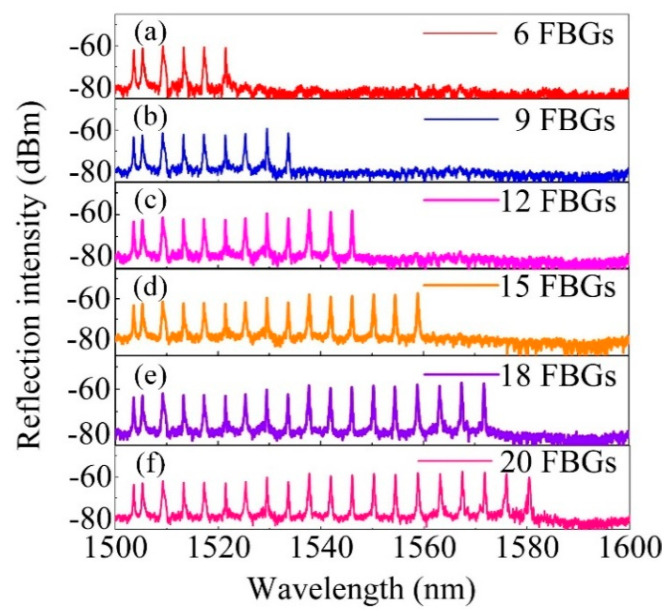
Reflection spectra evolution (**a**–**f**) of an FBG array inscribed in a H_2_-loaded SMF.

**Figure 8 sensors-20-04307-f008:**
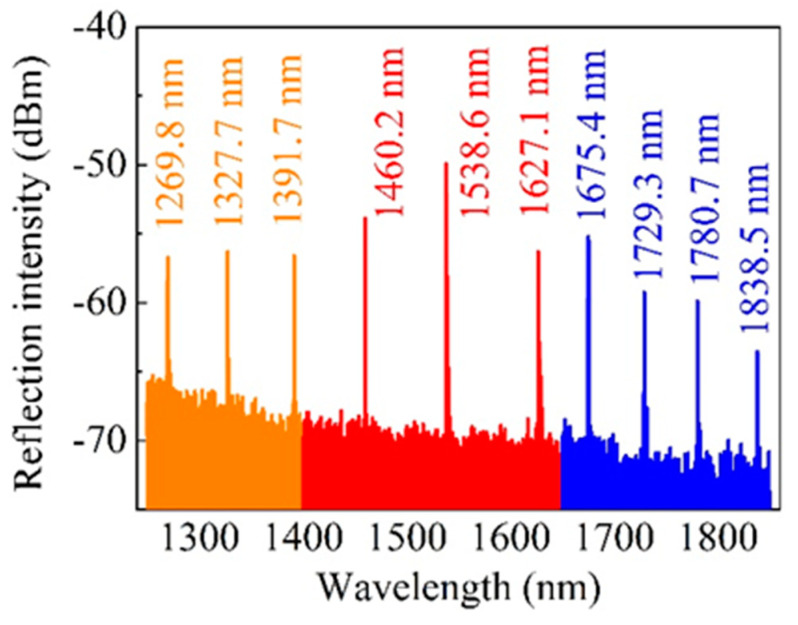
Reflection spectrum of an FBG array inscribed by the proposed Talbot interferometer without any compensating for the optical path difference (OPD). The central wavelength of inscribed FBG ranges from ~1269.8 to ~1838.5 nm.

**Table 1 sensors-20-04307-t001:** Calculated relationship between the rotation angle Φ of the mirror and the interference angle Φ’, central wavelength of fiber Bragg grating (FBG) Λ_FBG_, respectively.

Φ (°)	θ′ (°)	λ _FBG_ (nm)
5.00	24.46	928.94
2.00	18.46	1214.73
0.00	14.46	1540.37
−0.50	13.46	1652.46
−1.00	12.46	1782.72
−1.50	11.46	1935.92
−2.00	10.46	2118.63

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
