# Peer review of "High-Efficiency Inscription of Fiber Bragg Grating Array with High-Energy Nanosecond-Pulsed Laser Talbot Interferometer"

_sensors, 2020, doi:10.3390/s20154307_

Round 1
Reviewer 1 Report
The article describes practical implementation of Talbot interferometer and nanosecond-pulsed laser for inscription of fiber Bragg grating array. The idea is previously described in ref. [23], and the paper represents improvement of the method using the high-energy nanosecond-pulsed laser. Therefore, all the details of the improved method should be described. The missing parts are the following:
- The values of Wz and L_DOF should be given for the considered case (since all the parameters are given it will be great for readers to see these values for the considered interferometer).
- The grating length is estimated to be 6 mm and 8.3 mm long and the axial distance between gratings is 5mm (page 5). That means that there is overlapping of the neighboring gratings. A comment about that should be given.
- FBG array inscribed in H2-free SMFs (Figures 3 and 4): the time of exposure should be given.
- FBG array inscribed in H2-free SMFs: the weight of hanging mass block should be given.
- FBG array inscribed in H2-loaded SMFs: T1 and T2 curves are not explained (Figure 5.b)
- FBG array inscribed in H2-loaded SMFs: why the exposure time of each FBGs is 5 minutes (Figure 6) since the similar properties are obtained with 30 s exposure time (Figure 5)
Two small errors in text:
(a) Word where after eq. (4) should be with small letter w.
(b) Last eq. at page (4): instead of low it should be written lower.
Reviewer 2 Report
In this work, the authors present a Talbot interferometer for FBG inscription. The FBG inscription using Talbot interferometers is well established. The authors do not make it clear in the manuscript what is novelty in the technique presented. As the core of the work is the technique, a literature review on the use of Talbot interferometers to record FBG should be performed more carefully to highlight the contribution of this work. I am not convinced that Sensors is the right journal for this work, maybe the authors should choose a journal with scope in instrumentation. If the authors decide to resubmit the paper, I have some questions that should be addressed:
- The most important is to clearly demonstrate the novelty.
- The system should be better detailed. The authors superficially describe the system by mentioning that the mirrors and the support, in which the fiber is attached, can move. No details are provided to allow reproduction of the system.
- The manuscript has language issues and typos. The manuscript requires a careful revision.
Round 2
Reviewer 2 Report
In their revised version of the manuscript, the authors add a more detailed literature review highlighting the contribution of the work to the field. The minor questions pointed out in the first round were answered properly, so the manuscript is acceptable for publication.